# Anthocyanin Production from Plant Cell and Organ Cultures In Vitro

**DOI:** 10.3390/plants13010117

**Published:** 2023-12-31

**Authors:** Hosakatte Niranjana Murthy, Kadanthottu Sebastian Joseph, Kee Yoeup Paek, So-Young Park

**Affiliations:** 1Department of Botany, Karnatak University, Dharwad 580003, India; 2Department of Horticultural Science, Chungbuk National University, Cheongju 28644, Republic of Korea; 3Department of Life Sciences, CHRIST (Deemed to be University), Bengaluru 560029, India; ksjoseph15@gmail.com

**Keywords:** anthocyanins, colorants, elicitation, metabolic engineering, plant cell cultures, secondary metabolites

## Abstract

Anthocyanins are water-soluble pigments found in plants. They exist in various colors, including red, purple, and blue, and are utilized as natural colorants in the food and cosmetics industries. The pharmaceutical industry uses anthocyanins as therapeutic compounds because they have several medicinal qualities, including anti-obesity, anti-cancer, antidiabetic, neuroprotective, and cardioprotective effects. Anthocyanins are conventionally procured from colored fruits and vegetables and are utilized in the food, pharmaceutical, and cosmetic industries. However, the composition and concentration of anthocyanins from natural sources vary quantitively and qualitatively; therefore, plant cell and organ cultures have been explored for many decades to understand the production of these valuable compounds. A great deal of research has been carried out on plant cell cultures using varied methods, such as the selection of suitable cell lines, medium optimization, optimization culture conditions, precursor feeding, and elicitation for the production of anthocyanin pigments. In addition, metabolic engineering technologies have been applied for the hyperaccumulation of these compounds in varied plants, including tobacco and arabidopsis. In this review, we describe various strategies applied in plant cell and organ cultures for the production of anthocyanins.

## 1. Introduction

Anthocyanins are flavonoid compounds that contain water-soluble pigments in red, pink, purple, and blue colors. They are naturally found in plants, especially flowers, fruits, and tubers, such as red rose, blue rosemary, lavender, grapes, strawberries, raspberries, red plums, bilberries, blackberries, pomegranates, sweet cherry, eggplant, red onions, red cabbage, purple carrots, and others (Figure 1) [1,2]. Additionally, black grains and cereals, including black rice, black bean, and black soybean, and leaf vegetables, such as red amaranth, red spinach, and red leaf lettuce, are rich in anthocyanins; these could be used as a source of natural colorants and functional foods [3,4]. As natural colorants, anthocyanins have demonstrated many health benefits; they have antioxidant, anticancer, antidiabetic, anti-obesity, antimicrobial, neuroprotective, and cardioprotective properties, and can improve vision health; therefore, anthocyanins extracted from edible plants have been used in pharmaceuticals [5].

The anthocyanidins are made up of an aromatic ring [A] bound to an oxygen-containing heterocyclic ring [C], which is surrounded by a third aromatic ring through a carbon–carbon bond [B] [6] (Figure 2). The term “anthocyanins” refers to the glycoside form of anthocyanidins. Several anthocyanidins have been reported in plants; they differ in terms of the number and position of hydroxyl and/or methyl ether groups [7]. The predominant anthocyanidins found in plants are cyanidin, delphinidin, malvidin, pelargonidin, peonidin, and petunidin (Figure 3), which are present in 80% of pigmented leaves, 69% of pigmented fruits, and 50% of flowers [8]. In nature, these aglycones are attached to sugars and can potentially be further acylated by aliphatic or aromatic acids. The fact that simple anthocyanidins are rarely encountered in nature can be explained by the fact that both glycosylation and acylation increase anthocyanin stability [9].

The substitution pattern of the anthocyanidin’s B-ring, the degree of glycosylation, the type and degree of esterification of the saccharides with aliphatic and aromatic acids, as well as the pH, temperature, solvent, and presence of other pigments, are the primary factors influencing color variations among anthocyanins [7]. The pH dependence of anthocyanins’ color is one of their main traits (Figure 4). They have a red hue in an aqueous solution at pH 1–3, produce a colorless carbinol pseudo base at pH 5, are pink at pH 6–7, are blue-purple quinoidal bases at pH 7–8, and are pale yellow at pH 8–9. A number of enzymes, light, oxygen, metal ions, and other variables can potentially affect color alteration [1,7].

Since anthocyanins are naturally occurring food colorants with appealing properties, the food industry has long used them. Due to legislative limitations on some synthetic red dyes, the food industry is especially in need of water-soluble natural red colorants [10]. Anthocyanins’ relative volatility and interactions with other food matrix elements have hindered their commercial use as food colorants. The structure and concentration of anthocyanin, as well as pH, temperature, light, oxygen, solvents, and the presence of other flavonoids, proteins, and metallic ions, all have an impact on its stability [1]. This is why a major area of recent research has been the chemical stabilization of anthocyanins.

It has been demonstrated that acylation and co-pigmentation improve anthocyanin stability. The stability of anthocyanins can be increased by a number of techniques, including light preservation and oxygen exclusion. New food colorants with improved stability, bioavailability, and solubility have been made possible in recent years by the microencapsulation of anthocyanins [1,11].

Due to their many applications, anthocyanins are typically isolated from a variety of vegetables (like purple/black carrots) and fruit pulp (like grapes) and are used in the food, pharmaceutical, and cosmetic industries. However, cultivars, growing conditions, area, cultivation methods, maturity, processing, and storage all affect the anthocyanin content of fruits and vegetables [1]. Due to the conventional method’s inability to complete direct extraction from plant raw materials and meet the growing global demand for anthocyanins, significant research efforts have resulted in the development of more effective alternative plant in vitro cultures as well as the novel production of anthocyanins through the use of metabolic engineering [12,13]. We present here an overview of the different methods that have been discovered and applied to enhance anthocyanin synthesis through the use of plant in vitro cultures and metabolic engineering.

## 2. Biosynthesis of Anthocyanins

Anthocyanins have a 15-carbon base structure comprised of two phenyl rings (called A- and B-rings) connected by a three-carbon bridge that usually forms a third ring (called the C-ring) (Figure 2). Most anthocyanins are derived from just three basic anthocyanidin types: pelargonidin, cyanidin, and delphinidin. The precursor molecule for anthocyanin biosynthesis is phenylalanine; initially, phenylalanine is converted into 4-coumaroyl-coenzyme A, which is the key substrate for the biosynthesis of flavonoids, the synthesis of 4-coumaroyl-coenzyme A is catalyzed by the phenylalanine ammonia-lyase (PAL), cinnamate-4-hydroxylase (C4H) and 4-coumaroyl: CoA ligase (4CL) enzymes (Figure 5). Subsequently, there will be a synthesis of a chalcone molecule, usually naringenin-chalcone (2′,4,4′,6-tetrahydroxychalcone, THC), which is formed by the condensation of one molecule of 4-coumaroyl-CoA and three molecules of malonyl-CoA that are catalyzed by chalcone synthase (CHS). THC is an unstable molecule that is involved in isomerization for the formation of naringenin-flavanone; this reaction is catalyzed by chalcone isomerase (CHI). Naringenin-flavanone is the base molecule for the synthesis of flavones and flavanols. In the subsequent steps, there will be hydroxylation of the B ring of naringenin-flavanone, which is responsible for coloration, and an increase in hydroxylation, which causes a significant change in color from the extreme magenta to the blue end of the spectrum [14,15,16]. Hydroxylation of naringenin-flavanone is catalyzed by flavanone-3-hydroxylase (F3′H) and flavonoid 3′,5′-hydroxylases (F3′5′H); both are members of the Cytochrome P450 family (Cyt P450). Cyt P450 uses molecular oxygen and NADPH as cofactors and dihydroflavonol dihydrokaempferol (DHK) as a substrate to catalyze hydroxylation at 3- and 5- of the B ring to produce dihydroquercetin (DHQ, which is a precursor of cyanidin), dihydrokaempferol (DHK, which is the precursor of pelargonidin), and dihydromyricetin (DHM, which is the precursor of delphinidin), respectively (Figure 5). Both F3H and F3′5′H hydroxylases play a major role in the metabolism of flavonoids (Figure 5).

In the last stages of anthocyanidin formation, dihydroflavonol 4-reductase (DFR) enzyme converts dihydromyrcetin (DHM), dihydroquercetin (DHQ), and dihydrokaempferol (DHK) to produce the corresponding luecoanthocyanidins (Figure 5). Another enzyme, anthocyanidin synthase (ANS), in the last step reactions, catalyzes the conversion leucoanthocyanidins into cyanidin, pelarogondin, and delphnidin which are colored compounds. The synthesized anthocyanidins are then modified by a series of glycosylations and methylations that are catalyzed by UPD-glucose, flavonoid-3-glucosyl transferase (UFGT), and methyltransferase (MT), to form stable anthocyanins [15,17,18,19].

## 3. Production of Anthocyanins Using Plant Cell and Organ Cultures

For five to six decades, plant cell cultures have been studied to understand their part in in vitro production of anthocyanins for use in food, nutraceutical, pharmaceutical, and cosmetic industries, and cell or organ cultures have been initiated in more than 50 plant species [20]. Researchers experienced pigmentation in cell cultures regardless of the plant, species, source of explants, and types of cultures established since the flavonoid biosynthetic pathway is common to all flowering plants. Three significant plant species—carrot (*Daucus carota*), grape (*Vitis vinifera*), and strawberry (*Fragaria ananassa*)—have been the subject of in-depth research on in vitro cell cultures for the production of anthocyanins, among others. Several excellent review articles have been published and innumerable patents have been granted from time to time on anthocyanin production from in vitro cultures [20,21,22,23,24]. The flavonoid biosynthetic pathway is generally well-studied and established. In recent times, scientists have been working on genetic engineering and production-engineered plants and cell lines by cloning the regulatory genes and pathway genes involved in the anthocyanin biosynthetic pathway. They have also been using RNA interference to control the flavonoid flux in experimental plants like *Nicotiana tabacum* and *Arabidopsis thaliana* in order to produce anthocyanins for commercial use [12,24]. In the current review, we attempted to generalize the plant cell culture techniques and strategies used for a hyperaccumulation of anthocyanins by using well-studied systems like grape, carrot, and strawberry as well as model systems, like tobacco and arabidopsis, despite the difficulty of summarizing extensive research and comparing one cell culture system with others.

### 3.1. Strategies Applied for the Production of Anthocyanins in Cell and Organ Cultures

In many species, adventitious root, callus, cell, and hairy root cultures have been established (Table 1); strategies for biomass and anthocyanin production have been applied, including line/clone improvement or callus/cell/organ lines selection, culture medium selection, optimization of nutrient medium, phytohormones, and optimization of physical factors like light, temperature, hydrogen ion concentration (pH), precursors, and elicitors. In the section that follows, these strategies have been explained with appropriate examples.

#### 3.1.1. Selection of Cell Lines

It has been shown that the capacity of plant cells to produce secondary metabolites varies significantly. The majority of cell culture research begins with the selection of a high-quality cell line for the accumulation of secondary metabolites [70]. To develop high anthocyanin-yielding cell lines, for instance, cell line selection for *Euphorbia milli* was done over numerous callus subcultures, choosing cells with increased anthocyanin content in each subculture [22]. Two cultivars of grapes, *Vitis* hybrid Baily Alicant A and *V. vinifera* cv. Gamy Freaux, have been routinely utilized to establish cell suspension cultures [71]. Excellent results have been obtained once again through the careful selection of cell clusters over repeated sub-cultures of higher-performing cells [71,72]. A further illustration of the significance of effective lines is provided by the hairy root cultures of black carrots. Barba-Espin et al. [41] used wild-type *Rhizobium rhizogenes* to produce 93 lines of hairy roots in black carrots. Three fast-growing hairy root lines were chosen, two of which were from root explants (NB-R and 43-R lines) and one from a hypocotyl explant (43-H line). They showed that, in comparison to the original carrot line, the chosen lines produced 25–30 times more biomass and nine distinct anthocyanins.

#### 3.1.2. Optimization of Nutrient Medium

Establishing cell and organ cultures for the synthesis of plant secondary metabolites requires careful screening and the selection of appropriate media [70]. Different media formulations, with varying modifications, have been tested for establishing cell and organ cultures in various species for the production of anthocyanin (Table 1). These include Gamborg’s B5 medium [73], Linsmaier and Skoog or LS medium [74], Murashige and Skoog or MS medium [75], Shenk and Hildebrand or SH medium [76], and Woody plant medium or WPM [77]. In general, cell suspension cultures of *Angelica archangelica*, *Aralia cordata*, *Cleome rosea*, *Daucus carota*, *Melostoma malabathricum*, *Oxalis linearis*, *Panax sikkimensis*, and *Prunus cersus* could be established using MS media. It was discovered that LS medium was appropriate for cultivating cell cultures of *Perilla frutesens* and *Fragaria annanasa*; B5 medium was found to be appropriate for *Vaccinium macrocorpon* and *Vitis vinifera*; and WPM medium was found to be beneficial for *Ajuga pyramidalis* cell cultures (Table 1). The physiological state of the plant species and the kind of culture determine whether a certain medium is appropriate for a given species [78,79,80,81]. When Narayan et al. [39] tested B5, LS, MS, and SH media for their ability to grow *Daucus carota* cell cultures, they found that MS medium produced the most biomass and anthocyanin synthesis, while SH medium supported biomass and inhibited anthocyanin synthesis, and other media had lower levels of both biomass and anthocyanin. In hairy root cultures of carrots, Barba-Espin et al. [41] investigated the effects of 1/4, 1/2, and full-strength MS medium on biomass and anthocyanin accumulation. They found that, while full and 1/2 MS were beneficial for biomass accumulation, 1/2 MS was responsible for 4- to 6-fold higher anthocyanin production in various hairy root lines than full MS. They recommended a 1/2 MS medium for the accumulation of biomass and anthocyanin.

#### 3.1.3. Influence of Carbon, Nitrogen, and Phosphorous

Optimization of medium, especially with respect to carbon, nitrogen, and phosphorous, has shown a significant impact on the growth of cultured cells and anthocyanin synthesis. The type and concentration of sugars in the medium have demonstrated a profound influence on the growth and synthesis of anthocyanins in cell cultures of varied species (Table 1). It was demonstrated with grape cell cultures that simple sugars, such as glucose, galactose, and sucrose, or metabolizable sugars support the growth of the cells and accumulation of biomass, whereas non-metabolizable sugars, such as mannitol, are responsible for osmotic stress, which triggers the accumulation of anthocyanins [82,83]. Rajendran et al. [84] subjected carrot callus cultures to high concentrations of both sucrose and mannitol and showed that such conditions resulted in an increase in anthocyanin production because of enhanced osmotic conditions. Such carbon effects have been also demonstrated in *Ajuga pyramidalis* [25], *Cleome rosea* [30], and *Rosa hybrida* [58]. In more recent studies, Dai et al. [85] displayed the effect of sugars (glucose, fructose, and sucrose) on in vitro cultured grape berries (in vitro culturing of intact detached grape berries that were grown in greenhouse conditions); their work showed that glucose, fructose, and sucrose increased anthocyanin accumulation, with glucose and fructose being more effective than sucrose. Through molecular analysis, Dai et al. [85] illustrated that sugar induced enhanced anthocyanin accumulation in in vitro-grown grape berries, which resulted from altered expression of regulatory and structural genes, including *CHS*, *CHI*, *F3′H*, *F3H*, *DFR*, *LAR*, *LDOX*, and *ANR*.

Plant cell culture medium consists of both nitrate and ammonium forms of nitrogen, and the concentration of these two types of nitrogen has been shown to have profound influence on the growth of biomass and anthocyanin synthesis. It has been shown that the reduction of the ammonium form of nitrogen in the medium and enhancement of nitrate nitrogen favored cell growth and anthocyanin accumulation in many types of cell cultures (Table 1). The concentration of total nitrogen is 60 mM, and the ratio of NH_4_^+^ and NO_3_^−^ is 1: 2 in MS medium; when the ratio of NH_4_^+^ to NO_3_^-^ was varied from 1:1 to 1:32, keeping nitrogen concentration at 60 mM, there was increased production of anthocyanin (6.8% increment) in carrot cell suspension cultures [37]. However, they recorded optimal cell growth with the medium containing a 1:2 ratio of NH_4_^+^ to NO_3_^−^; therefore, this medium was responsible for a reduction in anthocyanin production of 3.5%. It was reported that cell growth and anthocyanin production was maximized when the ratio of NH_4_^+^ to NO_3_^−^ was 1:1 at 60 mM in grape [86], 1:16 at 30 mM in Christ plant [87], 2:28 at 30 mM in strawberry [43], and 1:16 at 30 mM in spikenard [88] cell cultures. These results indicate that the reduction of ammonium nitrogen and increment in nitrate nitrogen favors anthocyanin synthesis in cell cultures of several plants. Another contrasting study by Hirasuna et al. [60] showed that the limitation of overall nitrogen concentration in the culture medium and the increment in sugar levels resulted in a ‘switch-like’ (rather than gradual) enhancement of anthocyanin production in grape cell cultures, which may be equivalent to the removal of an inhibitory effect. The possible explanation given for this short phenomenon is in line with the result which stated that elevated sugars are responsible for increment osmotic potential and concomitant reduction in cell division; this makes nutrient resources more available for secondary metabolism, leading to enhanced biosynthesis of anthocyanin. In contrast, the opposite situation is true for cell growth when higher concentrations of nitrogen levels and appropriate concentration of sugars. Recent studies by Saad et al. [40] have also shown a similar phenomenon in carrot cell suspension cultures. They demonstrated that an altered concentration of NH_4_NO_3_ and KNO_3_ (20:37.6 mM) of MS medium affected the transcription levels of anthocyanin biosynthetic genes, including *PAL*, *4CL*, *CHS*, *CHI*, *LDOX*, and *UFGT*, which is accountable for the increased concentration of anthocyanin content.

Another element that has been demonstrated to have a significant impact on the synthesis of anthocyanins in *Vitis vinifera* [63] and *Daucus carota* [84] is the phosphorus level in the cell culture medium. In *Vitis vinifera* cell cultures, phosphate levels were reduced from 1.1 mM to 0.25 mM, and without phosphorus. This resulted in increased anthocyanin synthesis by 32% and 46%, respectively [63]. During the culture period, they observed a simultaneous rise in dihydroflavonol reductase activity. Yin et al. [69] investigated the impact of phosphate deficiency on the biosynthesis of anthocyanins in *Vitis vinifera* cv. Baily Alicante in a different study. Increases in the expression of transcription factor-encoding gene *VvMybA1* and the flavonoid 3-O-glucosyl transferase (*UFGT*) gene are implicated in the pathway leading to anthocyanin production.

#### 3.1.4. Influence of Plant Growth Regulators

Plant cell cultures are normally supplemented with varied growth regulators, including auxins such as 2,4-D, IAA, IBA, and NAA, and cytokinins, such as BAP/BA, KN, 2-iP. In several cases, a combination of several auxins with cytokinins has been tested and used efficiently to increase the cell biomass and anthocyanin production. Less frequently, researchers have tested the impact of gibberellic acid (GA_3_) and abscisic acid (ABA) in cell cultures of some species. Overall comparative analysis of the impact of growth regulators reveals that the effect of auxins and cytokinins varies from species to species. Among the varied auxins tested, 2,4-D showed promotive effects at lower concentrations, triggering cell growth; however, at elevated concentrations it was found to be inhibitory for anthocyanin production (Table 1). In carrot (*Daucus carota* cv. Kurodagosun) cell suspension cultures, Ozeki and Komamine [33] initially demonstrated that anthocyanin formation was induced by transferring the cells from 2,4-D containing medium to 2,4-D lacking medium. In subsequent studies, Ozeki et al. [89] showed that the carrot cell suspension involved in activities of phenyl ammonia-lyase (PAL), chalcone synthase (CHS), and chalcone-flavanone isomerase (CHFI) activities decreased when they were transferred to the cells from 2,4-D containing medium to 2,4-D lacking medium. Liu et al. [28] demonstrated the function of various auxin concentrations (0, 0.2, 0.4, 2.2, 9, 18, and 27 µM) in transgenic *Arabidopsis thaliana*, including IAA, NAA, and 2,4-D. Among transgenic *Arabidopsis thaliana*, these auxins variably regulated the expression of genes involved in anthocyanin biosynthesis, including four pathway genes (*PAL1*, *CHS*, *DFR*, and *ANS*) and six transcription factors (*TTG1*, *EGL3*, *MYBL2*, *TT8*, *GL3*, and *PAP1*).

Narayan et al. [39] studied the influence of 2,4-D, IAA, and NAA at different levels and recorded the decreased anthocyanin productivity with the increase in 2,4-D levels, and among the three auxins tested, they showed maximum biomass as well as anthocyanin production only when IAA was present in the medium. In addition, Narayan et al. [39] tested the influence of BAP, KN, and 2-iP (0.1–0.4 mg/L) in combination with IAA (2 mg/L), and again, a low level of KN (0.2 mg/L) showed the highest productivity of anthocyanin. Cytokinin’s effect on the gene expression involved in the process of anthocyanin biosynthesis, including *PAL*, *CHS*, *CHI*, and *DFR* genes, has been demonstrated by Deikman and Hammer [90]. The addition of KN encouraged the growth of biomass, whereas the substitution of BAP for KN reduced the formation of anthocyanins. In addition to auxins and cytokinins, Gagne et al. [65] have demonstrated the impact of the growth regulator ABA on the formation of anthocyanins in grape cell cultures. Research revealed that the expression of anthocyanin biosynthesis genes, such as *PAL*, *C4H*, *CHI1*, and *CHI2*, could be induced when ABA was supplemented in the medium. The findings presented above imply that choosing the right combination and concentration of plant growth regulators is essential for producing anthocyanins and cell biomass in plant cell cultures.

#### 3.1.5. Influence of Light, Temperature, and Medium pH

Varied factors, including light, temperature, and medium pH, are reported to be useful components that should be optimized before using the cultures to scale up the process, and these factors influence both biomass and secondary metabolite accumulation in plant cell and organ cultures [70,78]. Fluorescent light, when applied to callus or cell suspension cultures of carrot [91], grape [24], strawberry [92], perilla [53], senduduk [50], and cranberry [59], demonstrated a stimulatory effect on biosynthesis and accumulation of anthocyanins (Table 1). The positive effect of light in relation to PAL activity has been shown by Takeda et al. [91]. The influence of UV-A, UB-B, and UV-C lights has been demonstrated in grape berries; their effect is correlated with the stimulation of expression of the structural genes that encode the enzymes in the shikimate pathway [93].

Studies on *Perilla furtescens* [53], and *Melastoma malbarthricum* [50] have examined the effects of temperature regimes during in vitro cell cultures. Generally, relatively higher temperature levels (25 to 30 °C) favored cell growth and biomass accumulation, while relatively lower temperatures (20 °C) facilitated the synthesis of anthocyanins. Zhang et al. [94] used a two-stage culture strategy, maintaining the cultures at 30 °C for three days before moving them to 20 °C. This allowed them to achieve an optimal anthocyanin content of 270 g/L, which was higher than what they would have obtained with a constant temperature treatment. Research conducted by Yamane et al. [95] on the anthocyanin accumulation in berries of the *Vitis* hybrid (*V. labrusca* × *V. vinifera*) revealed that 20 °C is preferred above 30 °C for a higher anthocyanin accumulation. Although the exact mechanism underlying the temperature effect is yet unknown, Azuma et al. [96] demonstrated the differential expression of MYB-related transcription factors that are temperature-regulated using qRT-PCR study. Additional research is required in this area of study.

The pH or hydrogen ion concentration of the culture media is another crucial element that promotes cell growth and metabolite accumulation. Extreme pH values should be avoided; generally, cells growing in the medium may absorb nutrients efficiently and participate in growth, development, and metabolism at a pH of 5.8 [70]. Numerous groups have conducted research on the effects of medium pH on cell proliferation and anthocyanin production. In *Vitis* hybrid Baily Alicant A, Suzuki et al. [97] examined cell growth, biomass accumulation, and anthocyanin synthesis in media with varying pH levels, ranging from 4.5 to 8.5. They found that the medium with a pH of 4.5 produced better cell growth and anthocyanin synthesis. Iercan and Nedelea [98] observed a comparable phenomenon in callus cultures of *Vitis vinifera* cultivars. The callus cultures of *Feteasca neagra* with the highest anthocyanin content (13.5 mg/g FW) were found on the medium with pH = 4.5, whereas the medium with pH = 7.5 revealed 3.3 mg/g FW of anthocyanin. Hagendoorn et al. [99] showed that, in a number of plant species, such as *Morinda citrifolia*, *Petunia hybrida*, and *Linum flavum*, cytoplasmic acidification promoted the formation of secondary metabolites. They demonstrated a rise in PAL activity in response to the culture medium’s cytoplasmic acidification. Numerous studies have shown that higher medium sugar concentrations and nitrate-to-ammonium ratios promote the accumulation of anthocyanins in cell cultures of different species. These characteristics are also responsible for the medium’s pH reduction and the acidification of cultured cells (Table 1).

#### 3.1.6. Elicitation

Elicitors are molecules of biotic and abiotic origin or physical stimuli, such as pulsed electric field or UV irradiation, that can trigger the biosynthesis of secondary metabolites in plant cell and organ cultures [100]. Elicitors such as methyl jasmonate, jasmonic acid, and salicylic acid have been used efficiently in plant in-vitro cultures to activate the biosynthesis of anthocyanins (Table 1). The extracts of bacterial and fungal origin; insect saliva/varied components of insect saliva; complex polysaccharides, such as chitosan, pectin, alginate, and cyclodextrin; high concentration varied salts, such as CaCl_2_, MnSO_4_, ZnSO_4_, CoCl_2_, FeSO_4_, VoSO_4_, CuSO_4_, NH_4_NO_3_, and KnO_3_; ethephon (ethylene producer); UV irradiation; and pulsed electric field stimulus have been tested as elicitors in cell and organ cultures to enhance the anthocyanin biosynthetic pathway (Table 1).

Treatment of cell suspension cultures of *Vitis vinifera* with methyl jasmonate (MeJA) has triggered a 2.9 to 4.1-fold increment in anthocyanin content [64,101]. Similarly, MeJA has been shown to enhance the anthocyanin accumulation in callus cultures of *Malus sieversii* [49]. However, the efficiency of the elicitation process depends on the plant material, culture conditions, contact time, and elicitor concentration. In another study, Qu et al. [64] tested the combined effect of precursor feeding (phenylalanine) treatment with MeJA treatment in grape cell cultures, and they achieved a 4.6-fold increment in anthocyanin accumulation with the addition of 5 mg/L phenylalanine and 50 mg/L MeJA. Sun et al. [49] showed that expression of anthocyanin regulatory (*MdMYB3*, *MdMYBB9*, and *MdMYB10*) and structural (*MdCHS*, *MdDFR*, *MdF3H*, and *MdUFGT*) genes increased in *Malus sieversii* in response to MeJA elicitation. In addition, Wang et al. [102] demonstrated the upregulation of *MdMYB24L* gene in response to MeJA treatment in apple cell cultures, which positively regulates the transcription of *MdDFR* and *MdUGFGT* genes. The molecular mechanism of JA elicitation is also in line with MeJA action, and Shan et al. [103] deciphered the role of F-box protein CO11 in regulating the transcription factors *PAP1*, *PAP2*, and *GL3*, which were responsible for the expression of anthocyanin biosynthetic genes *DFR*, *LDOX*, and *UF3GR* in *Arabidopsis thaliana*.

## 4. Application of Metabolic Engineering

The biosynthetic pathway involved in the production of anthocyanins has been thoroughly investigated, and the genes involved in the conversion of the precursors to final products have been elucidated [104,105]. The anthocyanin biosynthesis pathway genes have been cloned into *Nicotiana tabacum* and *Arabidopsis thaliana*. Strategies used during metabolic engineering to improve anthocyanin production include: (1) cloning regulatory genes that control the pathway genes, (2) cloning pathway genes involved in various steps of anthocyanin biosynthesis, (3) utilizing RNA interference (RNAi) for gene silencing, and (4) utilizing the clustered regulatory interspaced short palindromic repeats (CRISPR)/Cas system for gene knockout [12,23,104,105,106,107,108].

Transcription factors (TF) of anthocyanin biosynthesis genes, such as MYB (myeloblastosis family), bHLH (basic helix-loop-helix family), and WD40/WDR (beta-transducin repeat family), have been suggested to regulate anthocyanin production in plants. For example, *MdMYB1* in *Malus domestica*, R2R3-MYBs like *PAP1*, *PAP2*, *MYB113*, and *MYB114* in *Arabidopsis thaliana*, and *VvMYBA1* and *VvMYBA2* in *Vitis vinifera* all regulate the production of anthocyanin [104]. Liu et al. [109] proposed that the transcription of many structural genes is regulated by the MYB, bHLH, and WD40 (MBW) complex, which binds to and activates their promoters. Here, we are highlighting the research carried out by Appelhagen et al. [12], who have cloned the two MYB transcription factors from *Antirrhinum majus* namely MYB Rosea1 (*AmRos1*) and bHLH Delila (*AmDel*) to *Nicotiana tabacum* cv. Sumsun. They raised calli and cell suspension cultures generated from *AmDel*/*AmRos1* plants and reported the accumulation of 30 mg/g DW of cyanidin 3-O-rutinoside (C3R) with such cultures. They maintained the engineered tobacco *AmDel*/*AmRos1* cultures for more than 10 years without substantial reduction in anthocyanin production, demonstrating its stability without aging or silencing of the transgenes.

In another set of experiments, Appelhagen et al. [12] expressed a pathway gene encoding flavonoid 3′,5′-hydroxylase from *Petunia* × *hybrida* (*PhF3′5′H*) or a gene encoding an anthocyanin 3-*O*-rutinoside-4″hydoxycinnmoyl transferase from *Solanum lycopersicum* (*Sl3AT*) in *AmDel*/*AmRos1* tobacco lines. They demonstrated that the cell lines that were expressing *AmDel*/*AmRos1* and *PhF3′5′H* were purple due to the production of delphinidin 3-*O*-rutinoside (D3R) in addition to C3R. Appelhagen et al. [12] displayed that the engineered tobacco lines *AmDel*/*AmRos1*-*PhF3′5′H* and *AmDel*/*AmRos1*-*Sl3AT* were quite stable and produced 4-fold higher (20 mg/g DW) anthocyanins (C3R) when compared to wild grape cell cultures (5 mg/g DW). Through high-performance liquid chromatography analysis, Appelhagen et al. [12] deciphered that the anthocyanin extract of the *AmDel*/*AmRos1* line contained almost exclusively C3R, with minor amounts of cyanidin 3-*O*-glucoside (C3G), pelargonidin 3-*O*-rutinoside (Pel3R), and peonidin 3-*O*-rutinoside (Phe3R). They recorded similar profiles with *AmDel*/*AmRos1*-*PhF3′5′H* except for the additional production of delphinidin 3-*O*-rutinoside (D3R). Meanwhile, they reported aromatically acylated compounds with coumaroyl- or feruloyl-moieties, with cyanidin 3-*O*-(coumaroyl) rutinoside (C3couR) and cyanidin 3-*O*-(feruloyl) rutinoside (C3ferR) with *AmDel/AmRos1- Sl3AT* line. Similar to these findings, Shi and Xie [110] generated plants and cell lines in *Arabidopsis thaliana* by cloning the *pap1*-*D* gene. Mono et al. [111] demonstrated the overexpression of the *IBMYB1* gene in *Ipomea batatus* calli.

Nakatsuka et al. [112] have used an RNAi-mediated gene silencing technique to generate red-flower transgenic tobacco by suppressing two endogenous genes (*FLS*, *F3′H*) along with overexpression of a foreign DFR gene in gerbera, which shits the flux toward pelargonidin synthesis. Khusnutdinov et al. [105] discussed varied examples of the phenotypic effect of CRISPR/Cas in editing the *pap1*, *DFR*, *F3H*, and *F3′H* genes of several distinct plant species.

## 5. Scale-Up Process

Numerous species, including *Aralia cordata*, *Daucus carota*, *Nicotiana tabacum*, *Perilla frutescens*, *Vaccinium pahale*, and *Vitis vinifera* (Table 2), have been the subject of studies aimed at scale-up cell or callus suspension cultures, and a number of bioprocess parameters have been optimized [113,114,115,116]. Kobayashi et al. [113] used *Aralia cordata* cell suspension cultures to evaluate a 500-L pilot-scale bioreactor culture for the synthesis of anthocyanins. After 16 days of cultivation, and with the provision of a carbon dioxide supply in the culture vessel, they obtained a fresh weight cell yield of 69.2 kg, an anthocyanin yield of 545 g, and an anthocyanin content of 17.2% in the dried cells. Although these experiments have been conducted in bioreactors ranging in size from small to large, the technology of bioreactors has not been applied further for the synthesis of anthocyanins. The selection of species with a higher anthocyanin content and the optimization of all bioprocess parameters, such as nutrient medium, physical characteristics, and other parameters that drive biomass growth and pigment accumulation, are prerequisites for successful bioreactor applications for the production of pigments. Highly promising recent investigations were conducted by Appelhagen et al. [12], who employed a transformed tobacco line (*AmDel**/*AmRos1*) that was cultivated in 2 L stirred tank commercial fermenters using LS media containing 1 mg/L 2,4-D and 100 mg/L kanamycin for 14 days at 23 °C. They were able to achieve 180 mg per bioreactor run and 90 mg/L of cyanidin 3-*O*-rotinoside (C3G) with such refined procedures.

## 6. Extraction of Anthocyanins

Anthocyanins from plant sources have been extracted using a variety of conventional and sophisticated techniques. Anthocyanins are typically extracted using solvent extraction, which generally involves the use of polar solvents, such as methanol, ethanol, acetone, water, and alcohol [117]. Maximum recovery of anthocyanins from plant tissues has generally been found to be possible with polar solvents at concentrations of 60–80%. Choosing appropriate extraction solvents is essential because anthocyanins are extremely reactive compounds [118]. The extraction of anthocyanins from plant material has typically involved the use of acidified solvents; modest acetic acid and hydrochloric acid concentrations have been shown to increase the extraction yield. However, using acids with higher concentrations should be avoided as this could cause the glycosidic linkages to partially hydrolyze [118]. Typically, 1–2% of an acidic agent, such as citric acid, formic acid, or phosphoric acid, is utilized together with organic solvents, such as methanol, ethanol, and acetonitrile. It was recommended that GRAS solvent be used to prevent the health concerns associated with different organic solvents [117]. As a result, several researchers have adhered to the two-phase aqueous extraction method [119,120]. As an alternative, scientists have experimented with high hydrostatic pressure [121], pulsed electric field extraction [122], microwave-assisted extraction [123], pressurized liquid extraction [124], and supercritical CO_2_ extraction [125]. All of these methods have been successfully used to extract anthocyanins from various plant materials. In conclusion, based on the source of material, appropriate processes that are effective, safer, and can adopt reduced solvent usage and higher extraction yield could be utilized.

## 7. Conclusions

A growing number of food and cosmetic companies are using anthocyanins as natural colorants since they offer a number of advantageous dietary and therapeutic benefits. In order to produce anthocyanins for use as natural colorants in the food and cosmetic industries, researchers are focusing on plant cell and organ cultures as a viable alternative due to the limited availability and variability of anthocyanins’ quality and quantity from natural resources, like fruits and vegetables. Despite five or six decades of study on plant cell and organ cultures, commercial production of anthocyanin has not proven successful. However, anthocyanins may now be obtained in greater amounts because of advancements in metabolic engineering and the cloning of regulator and pathway genes in model plants like Arabidopsis and tobacco. Notwithstanding the abundance of studies conducted in the field of plant cell and organ cultures, appropriate plant system selection, metabolic engineering implementation, and RNA interference, CRISPR/Cas technologies have the potential to generate valuable transgenic lines that exhibit elevated concentrations of particular anthocyanin compounds. It is necessary to conduct more studies in the following areas: cell line selection, culture condition optimization, elicitation, and bioprocess technology optimization utilizing bioreactor cultures. The development of extraction technologies and increasing the bioavailability of anthocyanins by encapsulation and other techniques are two more areas that require coordinated efforts. The application of plant cell and organ cultures to meet the rapidly expanding market demand for natural colorants in the food and cosmetic industries may be made possible by advancements in the aforementioned technologies.

## Figures and Tables

**Figure 1 plants-13-00117-f001:**
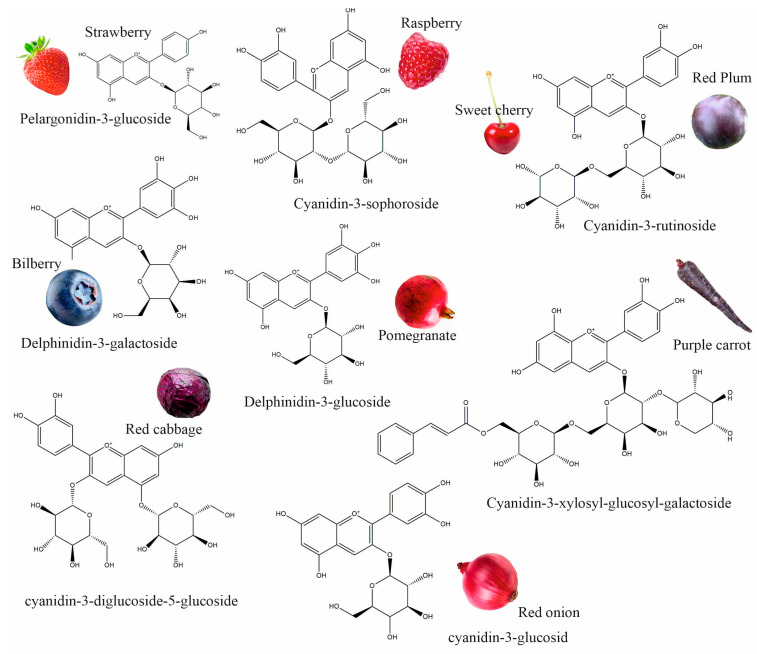
Different plant sources of anthocyanins.

**Figure 2 plants-13-00117-f002:**
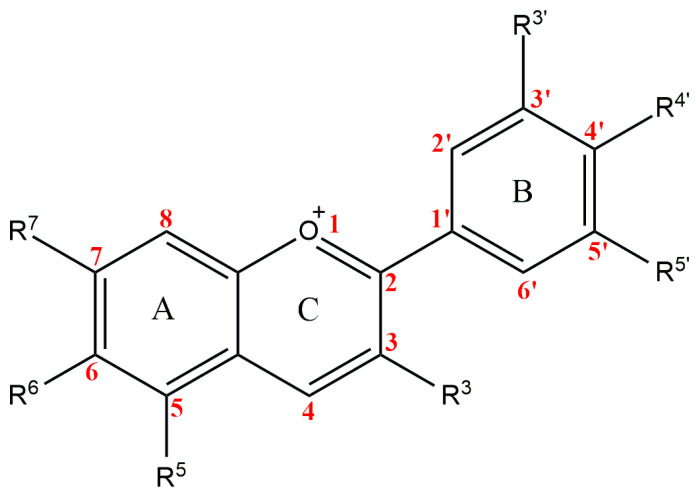
Basic anthocyanin structure.

**Figure 3 plants-13-00117-f003:**
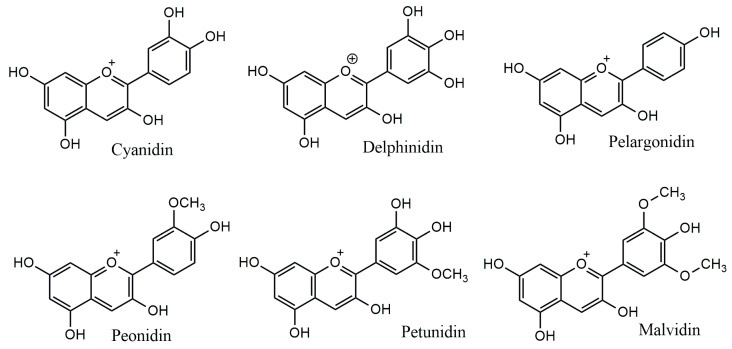
Major anthocyanidins from plants.

**Figure 4 plants-13-00117-f004:**
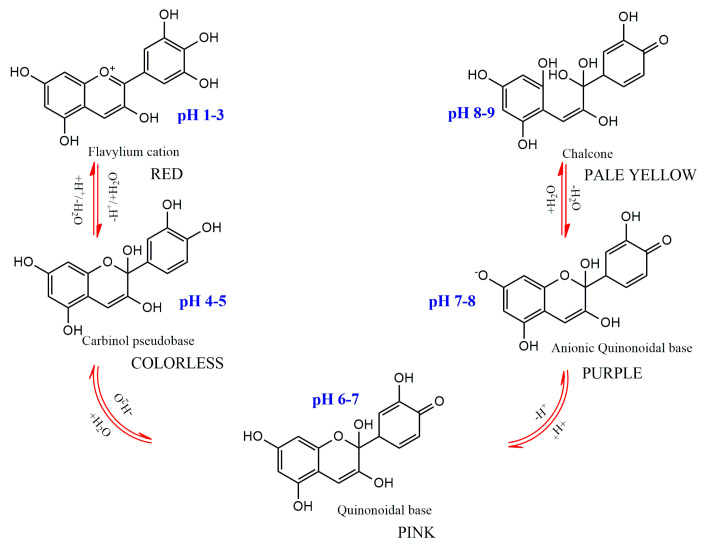
Structural transformations of anthocyanins in aqueous medium with different pH.

**Figure 5 plants-13-00117-f005:**
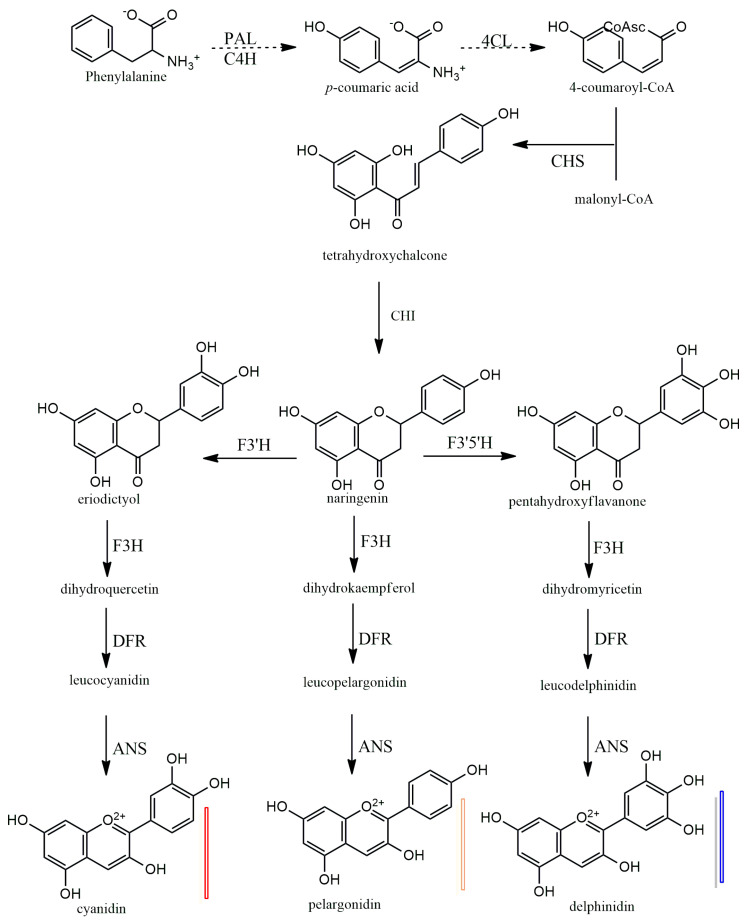
Biosynthesis of plant anthocyanidins. PAL—phenylalanine amino-lyase, C4H—cinnamate 4-hydroxylase, 4CL—4-coumarate: CoA ligase, CHS—chalcone synthase, CHI—chalcone isomerase, F3H—flavanone 3-hydroxylase, F3′H—flavonoid 3′-hydroxylase, F3′5′H—flavonoid 3′,5′-hydroxylase, DFR—dihydro flavanol 4-reductase, ANS—anthocyanidin synthase.

**Table 1 plants-13-00117-t001:** Successful examples of anthocyanin production from in vitro cell and organ cultures.

Plant Species	Type of Culture	Medium Composition	Strategy Followed	Response	Total Anthocyanin Content	References
*Ajuga pyramidalis* Metllica crispa	Cell suspension culture	Woody plant medium supplemented with 2.26 µM 2,4-D and 3.49 µM KN	The effect of different sugars was tested The effect of PGR was evaluated	Fructose in the medium resulted in the highest FW, whereas, galactose enhanced AP. 2,4-D plus kinetin resulted in higher growth, whereas a combination of IAA or NAA with zeatin resulted in the highest AP	0.29 mg/treatment	[25]
*Angelica archangelica*	Callus culture	MS medium with 30 g/L sucrose	The effect of PGR was evaluated	The highest biomass, of 231 mg DW, anthocyanin (11.22 mg/g DW), was achieved on a medium supplemented with 1 mg/L BAP	11.22 mg/g DW	[26]
*Arabidopsis thaliana*	Growth of seedlings after seed germination	MS medium with 3% sucrose	The effect of nitrogen in the MS medium was tested The effect of light was evaluated	The combined growth conditions of high light conditions induced the most molecular diversity and transcript levels of *PAP1*, *PAL1*, *CHS*, *DFR*, and *ANS* genes	NR	[27]
*Arabidopsis thaliana*	Callus culture	Transformed calli with *pap1-D* (production of anthocyanin pigmentation 1-Dominant) and wild-type calli were maintained on MS medium with 0.4 mM potassium nitrate (KNO_3_) and without ammonium nitrate (NH_4_NO_3_)	The effect of PGR was tested	2,4-D, NAA, and IAA control AP by regulating the expression of *TT8*, *GL3*, and *PAP1* (genes involved in regulating the gene expression) as well as *DFR* and *ANS* (genes involved in the anthocyanin biosynthetic pathway)	Anthocyanin content is expressed in relative units	[28]
*Aralia cordata*	Cell suspension culture	Varied media were tested	The effect of different media was tested	LS medium was good for biomass growth, however, B5 medium was good for AP	10.3% DW	[29]
*Cleome rosea*	Cell suspension culture	MS medium	The effect of salt strength on the medium was tested The effect of sucrose levels was evaluated	AP was highest in cell suspensions grown on half-strength MS, 30 g/L sucrose, and 0.45 µM 2,4-D	20.32 color value/g FW	[30]
*Daucus carota*	Callus culture	Modified MS medium with 0.1 mM 2,4-D and 60 mM sucrose	Selection of high-yielding cell line	Cultures of the smaller size had higher anthocyanin content than those of the larger-size	Anthocyanin content is expressed as relative absorbance units	[31]
*Daucus carota*	Callus culture	Modified MS medium with 0.1 mM 2,4-D and 60 mM sucrose	Selection of high-yielding cell line	Lines with higher levels of anthocyanin were selected	1–2%	[32]
*Daucus carota* cv. Kurodagosun	Cell suspension culture	MS medium containing 0.5 µM 2,4-D and 2% sucrose	The effect of 2,4-D was tested	The activity of varied enzymes involved in AP was suppressed with the addition of 2,4-D	NR	[33]
*Daucus carota*	Cell suspension culture	B5 medium with 0.1 mg/L 2,4-D and 100 mg/L MnSO_4_	The effect of different media was tested The effect of different sugars in the medium was tested	Of the varied media tested B5 medium resulted in the highest AP; Among the various carbohydrates, 20 g/L of galactose concentration resulted in the maximum cell yield and AP Of the varied combinations of sugars tested highest cell volume index and relative AP were observed in 15G:5S at 5% inoculum density	NR	[34]
*Dacus carota*	Callus culture	MS medium with 2 mg/L 2,4-D, 0.2 mg/L KN, 3% sucrose	The effect of fungal elicitors was evaluated	*Aspergillus flavus* extract at the 2.5% level resulted in the accumulation of a 2-fold increase in AP	20% DW	[35]
*Daucus carota*	Callus culture	MS medium with 2 mg/L 2,4-D, 0.2 mg/L KN, 3% sucrose	The effect of fungal extracts and metal ions were evaluated	All elicitors showed enhanced AP	NR	[36]
*Daucus carota* cv. Nantes scarlet 104	Callus culture	MS containing 2 mg/L 2,4-D, 0.2 mg/L KN	The effect of sugars and their concentrations was evaluated The effect of nitrogen levels were tested	Glucose and sucrose produced 3.5% (dry weight basis) AP, whereas supplementation of 7.5% sucrose to the medium produced the maximum AP (6.5%). Similarly, total nitrogen at 70 mM concentration and a 1:4 ratio of ammonium and nitrate yielded maximum callus growth and best AP	6.5% DW	[37]
*Daucus carota* var. Nantes scarlet	Callus culture	MS medium with 2 mg/L 2,4-D, 0.5 mg/L KN, 3% sucrose	The effect of MJ, and SA was tested	With the treatment of 200 µM SA and 0.01 µM MJ, an increment in AP was realized	0.36% and 0.37% (Control 0.22%)	[38]
*Daucus carota* cv. Nantes scarlet 104	Callus/cell suspension culture	The effect of MS/LS/B5/SH media was tested	The effect of different media and PGR were tested The effect of temperature was evaluated	Maximum biomass and AP were with MS medium. Among varied combinations of MS with IAA (4.0 mg/L) and KN (0.4 mg/L) was excellent for biomass and AP in liquid cultures. Of the varied temperature regimes 25 °C was excellent for both biomass and AP	2–4%	[39]
*Dacus carota*	Cell suspension culture	MS medium containing IAA (11.41 µM) and KN (0.93 µM)	The effect of nitrogen and phosphate levels was evaluated	The highest AP was obtained in the medium supplemented with 20.0:37.6 mM (NH_4_NO_3_:KNO_3_ ratio) on the 15th day. Similarly, a phosphate level of 0.45 mM in the medium supported the highest fresh cell weight (15.68 g/L) as compared to the culture supplemented with different concentrations of KH_2_PO_4_ (0, 0.225, 0.9, 1.8, or 2.7 mM), whereas maximum anthocyanin content was seen for 0.45 mM phosphate, which was 1.63-fold higher than in the control cultures	3.2 mg/g FW with the optimized nitrogen supplements and 3.66 mg/g FW with the optimized phosphate levels.	[40]
*Daucus carota* ssp. *sativus* var. *atrorubens*	Hairy root culture	Quarter, half, and full MS mediums were tested	The effect of medium strength was evaluated	Half-strength MS was suitable and 6-fold higher AP was accumulated compared to control	3.03 mg/g DW	[41]
*Dacus carota* var. Atomic red	Cell suspension culture	MS medium containing 9.1 µM IAA and 2.32 µM KN	The effect of salt stress was evaluated	The maximum AP was recorded in a salt stress medium of 4.33 mg/L FW on day 9 which was two-fold higher than the control	10.91 mg/L FW	[42]
*Fragaria ananassa* cv. Shikinari	Cell suspension culture	LS or B5 medium with 1 mg/L 2,4-D and 0.1 mg/L BA	The effect of various sugars was evaluated The effect of nitrogen concentrations were tested	The maximum yield of AP was realized in a modified LS medium containing 5% sucrose, a ratio of NH_4_^+^ (2 mM): NO_3_^−^ (28 mM)	380 µg/g FW	[43]
*Fragaria ananassa* cv. Shikinari	Cell suspension culture	LS medium supplemented with 3% sucrose, 0.1 mg/L BA, and 0, 0.01 and 1 mg/L 2,4-D	The effect of 2,4-D concentration was evaluated	Lower 2,4-D (0.1 mg/L) concentrations in the medium limited the cell growth and enhanced AP	1200 µg/g FW	[44]
*Fragaira annanassa* cv. Shikinari	Cell suspension culture	LS or B5 medium supplemented with 30 g/L sucrose, 1 mg/L 2,4-D and 0.1 mg/L BA	Repeated-batch culture strategy was evaluated on different media	The average AP was enhanced 1.7- and 1.76-fold by repeated-batch cultures in constant LS and constant B5 medium at a 9-day shift period for 45 days	0.42–0.52 mg/g FW	[45]
*Fragaria ananassa* cv. Shikinari	Cell suspension culture	LS medium with 1 mg/L 2,4-D, 0.1 mg/L BAP, 0.4 mg/L thiamine hydrochloride	Effect of precursor _L_-phenylalanine was evaluated	In repetitive feeding culture, a maximum of 81% higher AP was realized	58 mg/L	[46]
*Haplopappus gracilis*	Callus culture	Initial cultures were raised on a White medium with 2% sucrose and 10% coconut milk and 0, 0.1, 1.0, and 10 mg/L 2,4-D. During the second stage Sunderland medium with 2% sucrose and 10% coconut milk	The effect of 2,4-D, dark, and light incubation was evaluated	The light was essential for AP; however, 2,4-D suppressed the AP	3.9 mg/g DW	[47]
*Malus sieversii* f. *niedzwetzkyana*	Callus culture	MS medium with 3% sucrose	The effect of PGR levels and combination was evaluated	Auxin alone with the increment of concentration inhibited AP and molecular analysis showed that anthocyanin regulatory genes (*MdMYB10* and *MdbHLH3*) were dramatically suppressed by 0.6 mg/L 2,4-D. The BAP, TDZ, and nitrogen lower concentration of 2,4-D in combination with cytokinin enhanced the AP	Anthocyanin content is expressed as relative absorbance units	[48]
*Malus sieversii* f. *niedzwetzkyana*	Callus culture	MS medium with 4 µM BA, 2 µM NAA	The effect of MJ and ABA were tested	Studied the expression of gene expression and AP: Gene expression indicated that MJ induced expression of *MdMYB9*, *MdMYB10*, *Md CHS*, *MdF3H*, and *MdUFGT* genes. ABA inhibited the expression of *MdMYB3*, *MdMYB10*, *MDF3H*, *MdDFR*, *MdDOX*, and *MdUFGT* genes. Overall, 26% increment increase in cyanidin 3-O-galactoside content	Anthocyanin content is expressed as relative absorbance units	[49]
*Melostoma malabthricum*	Cell suspension culture	MS medium supplemented with 0.25 mg/L BA and 0.5 mg/L NAA	The effect of light intensity was evaluated The temperature levels were tested	The cultures exposed to light intensity of 301–600 lux depicted the highest biomass and AP; Cultures exposed to 20 °C have accumulated higher biomass and anthocyanin than 26 and 29 °C	1.62 color value/g FW	[50]
*Oxalis linearis*	Callus culture	MS medium supplemented with 60 mM sucrose	The effect of PGR was tested The effect of photoperiod was evaluated	Callus growth was stimulated at a concentration of 8–32 µM NAA and 2,4-D. BA and Zeatin inhibited callus growth, whereas IP stimulated callus growth. The optimum AP was with NAA at 8 µM NAA. Sucrose at 60 mM was best for callus growth and 240 mM was good for AP	NR	[51]
*Panax sikkimensis*	Callus culture	MS medium containing 3% sucrose, 0.01% myoinositol, 0.33 µM thiamine hydrochloride, 2.5 µM pyridoxine hydrochloride, 4.0 µM nicotinic acid, 4.5 µM 2,4-D, and 1.2 µM KN	Cell line selection The effect of photoperiod was evaluated	The selected line had the highest growth and productivity (2.192 mg/g FW and 6.92% DW); The continuous exposure to light favored both biomass and AP	2.76 mg/g FW (7.07% DW)	[52]
*Perilla frutescens*	Cell suspension culture	LS medium with 3% sucrose, 1 µM 2,4-D and 1 µM BA	The effect of temperature was evaluated	At temperatures of 22, 25, and 28 °C, the specific growth rate of cells was 0.21, 0.32, and 0.37 per day, respectively. However, the AP was reduced compared to 25 °C. Therefore, a temperate culture temperature of 25 °C was suggested for the maintenance of cultures	NR	[53]
*Perilla frutescens*	Cell suspension culture	LS medium with 3% sucrose, 1 µM 2,4-D and 1 µM BA	The effect of inoculum density was evaluated	Optimum AP (3.6 g/L) was reported at an inoculum size of 50 g/L	117.6 mg/g DW	[54]
*Perilla frutescens*	Cell suspension culture	MS medium supplemented with 0.2 mg/L 2,4-D and 0.5 mg/L BA	The effect of yeast extract was evaluated	Increment increase in AP (10.2%) with the addition of 1% yeast extract	10.2% DW	[55]
*Prunus cerasus* cv Amarena Mattarello	Callus culture	MS medium containing 30 g/L sucrose, 1 mg/L NAA, 0.1 mg/L BAP	The effect of JA was evaluated The light effect was tested	Cultures elicited by 50 µM JA stimulated the accumulation of cyanidin 3-glucoside accumulation When cultures were exposed to light, cyanidin 3-glucoside content was increased from 0.1 to 4.5 mg/100 g FW	5.3 mg/100 g FW	[56]
*Raphanus sativus* cv Peking Koushin	Adventitious root culture	Half strength MS	The effect of PGR was tested The effect of light was evaluated	Cultures with IBA IBA-supplemented medium and incubated in light involved in AP than cultures supplemented with NAA and dark incubation	0.15% DW	[57]
*Rosa hybrida*	Callus culture	B5 medium with 0.2 mg/L 2,4-D	The effect of sucrose concentration was tested The effect of nitrogen levels was evaluated	The optimum amount of AP (7.20 mg/kg FW and 6.58 mg/kg FW) was recorded on medium supplemented with 5% sucrose and medium devoid of NH_4_^+^ respectively	7.20 mg/kg FW	[58]
*Vaccinium macrocarpon*	Callus culture	Modified B5 medium with 5.7 µM NAA, 0.45 µM 2–4-D, and 2.32 µM KN in the dark at 25 °C	The effect of light was evaluated	Cultures exposed to light have shown maximum concentration of AP	140 µg/g FW	[59]
*Vitis* hybrid (Bailey Alicang A)	Cell suspension culture	B5 medium with 2% sucrose and 0.5 mg/L 2,4-D (Maintenance medium)	The combined effect of nitrogen and sucrose was evaluated	The combined effect of low nitrate and high sucrose synergistically improved AP	2.9 g/L	[60]
*Vitis vinifera* cv. Gamay Freaux	Cell suspension culture	B5 medium	The combined effect of nitrogen and sucrose was evaluated	The combined effect was responsible for the increase in anthocyanins, especially peonidin 3-glucoside	NR	[61]
*Vitis vinifera* var. Gamay Freaux	Callus culture	B5 medium with 88 mM sucrose, 250 mg/L casein hydrolysate, 0.54 µM NAA, 0.93 µM KN	Selection of cell line	Cell lines that were accumulating higher levels of peonidin 3-glucoside (line 5.4) and peonidin 3-p-coumaroylglucoside (line 13.1) were selected	1.02 mg/g FW	[62]
*Vitis vinifera* cv. Gamay Freaux	Cell suspension culture	Modified MS medium with B5 macro-elements, MS micro-elements, 2% sucrose, 0.025% casein hydrolysate, 0.1 mg/L KN and 0.1 mg/L NAA	The effect of phosphate level was tested	Deprivation of phosphate led to enhanced synthesis of anthocyanin by 32% and 46% in P2 and P3 media, respectively	NR	[63]
*Vitis vinifera* cv. Gamay Freaux var. Teinturier berry	Cell suspension culture	B5 medium with 30 g/L sucrose, 250 mg/L casein hydrolysate, 0.1 mg/L NAA, 0.2 µM KN	The effect of precursor phenylalanine and MJ was evaluated	Treatment with 5 mg/L phenylalanine, 50 mg/L MJ, and 1 mg/L dextran enhanced the 4.6-fold of AP	7.09 color value/g DW	[64]
*Vitis vanifera* cv Cabernet Sauvignon	Cell suspension culture	B5 medium supplemented with 20 g/L sucrose, 250 mg/L casein hydrolysate, 0.5 mg/L NAA, 0.12 mg/BA	The effect of ABA was evaluated	ABA-treated cells exhibited an earlier increase in *VvCHI1*, *VvCHI2, VvC4H,* and *VvMYBA1* transcripts and AP	NR	[65]
*Vitis vinifera* cv. Gamay Freaux	Cell suspension culture	B5 medium	The effect of indanoyl-isoleucine (In-Ile) was evaluated	In-Ile was a potent elicitor in stimulating AP and 2.6, 1.8, and 1.9-fold increments in anthocyanin production on 8 d, 10 d, and 12 d after the treatments, respectively	4.6 mg/g DW	[66]
*Vitis vinifera* cv. Gamay Freaux	Cell suspension culture	B5 medium 0.1 mg/L NAA, 0.2 mg/L KN, 0.25 mg/L casein hydrolysate, 3% sucrose	The effect of elicitors was evaluated	Chitosan, pectin, and alginate enhanced the production of anthocyanin by 2.5, 2.5, and 2.6-fold, respectively	4.2 mg/g DW	[67]
*Vitis vinifera* cv. Gamay Freaux	Cell suspension culture	B5 medium with 0.1 mg/L NAA, 0.2 mg/L KN, 0.25 mg/L casein hydrolysate, 3% sucrose	The effect of ethephon and pulsed electric field was evaluated	The treatment of ethephon resulted in a 2.3-fold increase (1.99 mg/g DW) and 2.3-fold (1.99 mg/g DW) in anthocyanin content, while combined treatment with both ethephon and PEF resulted in 2.5-fold increase (2.2 mg/g DW) in AP	2.2 mg/g DW	[68]
*Vitis* hybrid Bailey Alicante A	Cell suspension culture	MS medium supplemented with 3% sucrose, 0.23 µM 2–4-D, 1 µM KN, 3 µM thiamine-HCl, 560 µM myo-inositol	The effect of phosphate was evaluated	Cynidin-3-*O*-glucoside, peonidin-3-*O*-gluside, cyanidin-3-*O*-(6-*O*″-p-coumaroyl)-glucoside and peonidin-3-*O*-(6-*O*″-p-coumaroyl)-glucoside were detected in higher concentration in Pi deprived cells. The transcript levels of (*UFGT*) and *VvmbyA1* were also higher in Pi-deprived cells	NR	[69]

ABA—Abscisic acid; AP—Anthocyanin production; 2,4-D—2,4-dichlorophenoxy acetic acid; B5—Gamborg B5 medium; BA—benzyladenine; BAP—benzylaminopurine; DW—dry weight; FW—fresh weight; IAA—indole acetic acid; IBA—indole butyric acid; IP—isopentenyl adenine; JA—Jasmonic acid; KN—kinetin; LS—Linsmaier and Skoog medium; MJ—methyl jasmonate; MS—Murashige and Skoog medium; NAA—naphthalene acetic acid; NR-not reported; PGR—Plant growth regulator; SH—Schenk and Hildebrandt medium; TDZ—thidiazuron.

**Table 2 plants-13-00117-t002:** Successful examples of anthocyanin production from scale-up cultures.

Plant Species	Type of Culture	Medium Composition	Strategy Followed	Response	Total Anthocyanin Content	References
*Aralia cordata*	Cell suspension in 500-L jar fermenters (Stirred tank bioreactors)	MS medium with 1.0 mg/L 2,4-D, 0.1 mg KN and 30 g/L sucrose	The combined effect of airflow (0.2 vvm), CO_2_ (0.3%), and agitation (30 rpm) was evaluated	Anthocyanin content was 15.0% on a DW basis	17.2% DW (7.87 g/kg FW)	[113]
*Daucus carota*	Cell suspension cultures in 3.5-L (Stirred tank bioreactor)	MS medium with 11.4 µM IAA, 0.9 µM KN	The effect of salt stress (37.6 mM KnO_3_ and 20.0 mM NH_4_NO_3_) was tested	Enhanced AP was recorded	404 µg/g FW	[42]
*Nicotiana tabacum*	Cell culture in 2 L bioreactors (Stirred tank bioreactors)	LS medium supplemented with 1 mg/L 2,4-D and 100 mg/L kanamycin	Transgenic cell line containing *AmDel*/*AmRos1*	Enhanced AP was recorded	90 mg/L	[12]
*Perilla frutescens*	Cell suspension culture in 2-L capacity (Stirred tank bioreactor)	LS medium with 30 g/L sucrose, 10^−6^ M 2,4-D, and 10^−6^ M BA	The effect of oxygen transfer coefficient (*k_L_a*) was tested The effect of light irradiation was evaluated	With aeration of 0.2 vvm, the AP was 220 mg/g DW and 1.65 g/L The light irradiation enhanced a a two-fold increase in AP when compared to dark incubation	250 mg/g DW	[53]
*Vaccinium pahalae*	Cell suspension culture in 14-L(Stirred tank bioreactors)	Woody plant medium with 5.4 µM NAA, 0.45 µM 2,4-D, 20 µM BA, and 5% sucrose	The effect of agitation was evaluated	The effect of light was studied and cultures were exposed to light 40 µmol m^−2^ s^−1^ PPF and cultures were agitated at 150 rpm	75 mg/L	[114]
*Vitis vinifera* cell line VVG101	Cell suspension in 17-L column bioreactors (Stirred tank bioreactors)	B5 medium with 10 g/L sucrose, 250 mg casein hydrolysate, 0.1 mg/L NAA and 0.2 mg/L KN	The effect of the reciprocating plate system was tested on airflow conditions	Biomass accumulation was verified	NR	[115]
*Vitis vanifera* cv. Bailye alicant A	Callus culture in 500-mL bioreactors (Airlift bioreactors)	B5 medium with 30 g/L sucrose, 1 mg/L 2,4-D	The effect of osmoticum (0.8% carboxymethyl cellulose sodium salt) was tested	The average callus size was 490 µm, which was 1.6 times larger than that of conventional medium	33 mg/L	[116]

AP—Anthocyanin production; 2,4-D—2,4-dichlorophenoxy acetic acid; B5—Gamborg B5 medium; BA—benzyladenine; KN—kinetin; MS—Murashige and Skoog medium; NAA—naphthalene acetic acid; vvm—air volume/medium volume/minute.

## Data Availability

Not applicable.

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
