# Peer review of "Anthocyanin Production from Plant Cell and Organ Cultures In Vitro"

_plants, 2023, doi:10.3390/plants13010117_

Round 1
Reviewer 1 Report
Comments and Suggestions for Authors
Nevertheless the argument is not new, the review is quite interesting in summarizing different approaches to anthocyanin production.
English needs some revisions: some words are badly written and some sentences are not complete.
However, the paper is suitable for publication.

An extensive review of English is recommended.
Author Response
Response to reviewer comments
Manuscript ID: plants-2774518
Title: Anthocyanin production from plant cell and organ cultures in vitro
With reference to the above authors are thankful to anonymous reviewers for their valuable comments on the manuscript. We have revised the manuscript in the light of reviewer's comments and incorporated all the suggestions made by them. Following are the specific changes made in the revised manuscript.
Reviewer #1
Query #1 English needs some revision: some words are badly written and some sentences are incomplete
Answer: We have revised the manuscript in light of all three reviewers and the Assistant editor's comments. All the grammatical and typographical errors have been corrected and the revised manuscript was checked by a colleague who is a native English speaker.

Reviewer 2 Report
Comments and Suggestions for Authors
The review describe the importance of the anthocyanins in several field and the potential uses, the paper is correctly organised, well written, with a wide vision of the topic and with a great number of references consistent and updated, all cited correctly in the text. This review positioned to be a good support to the research in this field.
Author Response
Response to reviewer comments
Manuscript ID: plants-2774518
Title: Anthocyanin production from plant cell and organ cultures in vitro
With reference to the above authors are thankful to anonymous reviewers for their valuable comments on the manuscript. We have revised the manuscript in the light of reviewer's comments and incorporated all the suggestions made by them. Following are the specific changes made in the revised manuscript.
Reviewer #2
The revised manuscript is prepared as per the suggestions of other reviewers and the Assistant Editor.

Reviewer 3 Report
Comments and Suggestions for Authors
Dear Authors,
Generally, the manuscript is well written and comprehensive, based on the pertinent and broad literature.
I have some recommendations, listed below.
Page 6 line 154
Table 1 should be used to summaries the text. You need to give the most successful anthocyanin production strategy in column “Strategy followed”, not all treatments. In column “Response”, please used abbreviations /for example fresh weight FW, anthocyanin production AP etc./ The text in the Table is very detailed.
Page 19 line 174
What do you mean “original carrot strain”. The term “strain” is commonly used for bacteria.
The conclusion is well written. Summarizes success in the production of anthocyanins in plant cell and organ culture in vitro and specifies the opportunities for future studies.
Comments on the Quality of English Language
Minor editing of English language required.
Author Response
Response to reviewer comments
Manuscript ID: plants-2774518
Title: Anthocyanin production from plant cell and organ cultures in vitro
With reference to the above authors are thankful to anonymous reviewers for their valuable comments on the manuscript. We have revised the manuscript in the light of reviewer's comments and incorporated all the suggestions made by them. Following are the specific changes made in the revised manuscript.
Assistant Editor comments:
Query #1. I have some recommendations, listed below. Page 6 line 154, Table 1 should be used to summarize the text. You need to give the most successful examples…..
Answer: As per the suggestion of the reviewer and Assistant Editor – two tables are presented with the revised manuscript with suitable successful examples. Table 1 deals with successful examples of anthocyanin production from in vitro cell and organ cultures and Table 2 deals with successful examples of anthocyanin production from in scale-up cultures.
Query #2. Page 19 line 174
What do you mean by “Original carrot strain”….
Answer: It was an error and it has been corrected in the revised manuscript.

Reviewer 4 Report
Comments and Suggestions for Authors
The article authored by Murthy et al. review the advancements of various strategies toward the production of anthocyanin in in vitro cultures.
There is a related review published early this year (January 2023) also by MDPI at Molecules. I see the current work is aligned well with the recently published review, but still citation is missing. https://www.mdpi.com/1420-3049/28/2/866.
Table 1: please uniform the word Kinetin to be written in its abbreviated form 'KN' in the second paragraph 5th column.
The whole article is written in 29 pages, with this table occupying 13 pages. The table is made to give a good coverage of previously published systems, but very hard to follow. Here, I deeply recommend removing 'medium composition', 'strategy followed' and 'response' columns' from the main table. Please, shift the full current table 1 as it is to supplementary data, while leaving a footnote that more details can be found in supplementary table S1. In addition, any study related to scale-up process should be shifted to a separate main table 2 and cited within section 5 'Scale-up process'. While constructing new table 2, please consider to include the name of the bioreactor (ex. stirred tank, column, or airlift bioreactors…etc)
Line 286: "……. could be induced to express when ABA was supplemented in the medium." please revise the term induced to express.
One can re-form the whole sentence to:
Research revealed that the expression of anthocyanin biosynthesis genes such as PAL, C4H, CHI1, and CHI2 could be induced when ABA was supplemented in the medium.
Line 339: Replace Application with treatment.
Section 4 is very close to a recent review published earlier this year. Please consider citation and allow a better compromise. https://www.mdpi.com/1420-3049/28/2/866.
Introduction (line 84) and conclusions (line 433) are the only two positions where development of extraction conditions were briefly mentioned. I recommend having an additional section about the extraction of anthocyanins, where more researches can be discussed.
Comments on the Quality of English LanguagePlease consider shorter sentences, and typing error in the whole article.
Author Response
Response to reviewer comments
Manuscript ID: plants-2774518
Title: Anthocyanin production from plant cell and organ cultures in vitro
With reference to the above authors are thankful to anonymous reviewers for their valuable comments on the manuscript. We have revised the manuscript in the light of reviewer's comments and incorporated all the suggestions made by them. Following are the specific changes made in the revised manuscript.
Reviewer #4
Query #1 There is a related review published early this year (January 2023) also by MDPI at Molecules ….
Answer: The suggested article is cited in the revised manuscript in section 4.
Query #2 The whole article is written 29 pages, with this table occupying 13 pages….
Answer: As per the suggestion of the reviewer and Assistant Editor – two tables are presented with the revised manuscript. Table 1 deals with successful examples of anthocyanin production from in vitro cell and organ cultures and Table 2 deals with successful examples of anthocyanin production from in scale-up cultures.
Query #3 Line 286. ….. could be induced…
Answer: The line has been rewritten as per the suggestion.
Query #4. Line 339: Replace Application with treatment.
Answer: The suggestion is incorporated.
Query #5. Section 4 is very close to a recent review published earlier this year. Please consider citation….
Answer: The suggested article is cited in the revised manuscript in section 4.
Query #6. Introduction (line 84) and conclusion (line 433) are…. I recommend having an additional section about the extraction of anthocyanins…
Answer: The extraction of anthocyanins is presented in section 6 with a suitable discussion.
Query #7. Please consider shorter sentences, and typing errors in the whole article.
Answer: All the grammatical and typographical errors have been corrected and the revised manuscript was checked by a colleague who is a native English speaker.

Round 2
Reviewer 4 Report
Comments and Suggestions for Authors
The authors have improved the review article. I have no more points to address.